# Zwitterionic Functionalization of Persistent Luminescence Nanoparticles: Physicochemical Characterizations and In Vivo Biodistribution in Mice

Delphine Dassonville [1,†], Thomas Lécuyer [1,*,†], Johanne Seguin [1], Yohann Corvis [1], Jianhua Liu [1], Guanyu Cai [1], Julia Mouton [2,3], Daniel Scherman [1], Nathalie Mignet [1] and Cyrille Richard [1,*]

1    Université Paris Cité, CNRS, INSERM, Unité de Technologies Chimiques et Biologiques pour la Santé (UTCBS), 75006 Paris, France; delphine.dassonville@ens-lyon.fr (D.D.); johanne.seguin@u-paris.fr (J.S.); yohann.corvis@u-paris.fr (Y.C.); liujianhua37@163.com (J.L.); guanyu.cai@chimieparistech.psl.eu (G.C.); daniel.scherman@u-paris.fr (D.S.); nathalie.mignet@u-paris.fr (N.M.)
2    EPF Graduate School of Engineering, 34000 Montpellier, France; julia.mouton@epf.fr
3    Polymers Composites and Hybrids, IMT Mines d'Alès, 30100 Alès, France
*    Correspondence: thomas.lecuyer@alumni.chimie-paristech.fr (T.L.); cyrille.richard@u-paris.fr (C.R.)
†    These authors contributed equally to this work.

**Abstract:** After excitation in the biological transparency window, chromium-doped zinc gallate nanoparticles (ZGO NPs) emit near-infrared luminescence for more than an hour, allowing long-term imaging to be performed without background autofluorescence. However, these nanoparticles are recognized in just a few minutes by serum proteins and are then trapped in the liver. In this article, we put forth that liver uptake can be delayed when coating the surface of ZGO NPs with zwitterions. We focused on the use of a very small zwitterion molecule of 330 Da derived from sulfobetaine silane (SBS) and its grafting in one step and in water onto zinc gallate nanoparticles, and we compared the colloidal stability, the in vitro interactions with serum proteins, and the biodistribution in mice with PEGylated ZGO NPs (5000 Da) prepared in two steps in organic solvent. In vitro quantification of serum protein adsorption suggests that the similarity between the sulfobetaine and the cell membrane is enough to reduce protein adsorption as much as a PEGylation, despite the difference in coating thickness and molecular weight. This study has also proved that a combination of good protein repulsion and a smaller size compared to PEGylated NPs allows similar circulation times to be obtained in mice with zwitterionic or PEG coatings. Therefore, its use could offer new opportunities for further in vivo application of functionalized ZGO derivative NPs.

**Keywords:** nanoparticles; persistent luminescence; surface coating; zwitterion; nanomedicine

## 1. Introduction

Nanoparticles (NPs) are among the most promising candidates for biomedical imaging. Indeed, they can easily interact with similarly sized biological objects such as enzymes, membranes, or nucleic acids [1]. It allows them to be easily engineered to enter cells or cross the blood–brain barrier, overcoming the difficulty for organic fluorophores to target specific areas [2]. However, commonly used fluorescent nanoparticles suffer from critical drawbacks, such as tissue autofluorescence [3] during the continuous probe illumination and absorption of photons by components present in living organisms (water, hemoglobin, melanin, etc.) outside of the near-infrared range [4]. We previously reported the interest in chromium-doped zinc gallate nanoparticles (ZGOs) for biomedical imaging [5]. Their main characteristic is persistent luminescence (PLNPs), which means they continue emitting photons for several minutes if not hours after the end of excitation, unlike fluorescent probes [6–12]. This characteristic allows in vivo imaging to be performed without tissue autofluorescence, the maximum signal-to-noise ratio occurring when the acquisition is made a few seconds after the excitation source is turned off [13]. In addition to being

non-toxic [14–17] and having an infrared emission wavelength (696 nm) allowing deep tissue imaging [4], chromium-doped zinc gallate can be excited not only in the UV range but also near the biological transparency window. It is then possible to excite or re-excite ZGO nanoparticles in vivo with a simple LED and carry out the imaging experiment until they are eliminated or degraded by the body.

The main issue is that, when injected, ZGO nanoparticles are cleared from blood circulation in a matter of minutes and trapped in the liver and spleen [18]. Then, they cannot be monitored outside of these organs anymore. This stems from opsonization, a physiological mechanism whereby serum proteins called opsonins bind to exogenous species to enhance their elimination by phagocytes [19]. For some applications, it is necessary to reduce nanoparticle recognition by opsonins in order to extend their blood circulation time. This can be done by suppressing coulombic interactions using neutrally charged nanoparticles or by shielding the surface with highly hydrophilic groups to create a water barrier and prevent protein binding [20]. Studies also highlighted the key role of the hydrodynamic diameter of the NPs in the opsonization process [21]. To date, efforts have mainly involved the functionalization of the surface with polyethylene glycol (PEG), a neutral and hydrophilic polymer [22–24]. Notably, this ligand has been successfully grafted on ZGO nanoparticles in our group [5,18], as well as on other PLNPs [25–27]. The PEG functionalization of ZGO nanoparticles delays accumulation in the liver, which starts about six hours after injection in mice, whereas uncoated ZGO nanoparticles are trapped for a few minutes.

However, there are several drawbacks to the use of PEGylated nanoparticles. First, the polymers greatly increase the hydrodynamic diameter of nanoparticles, which might prevent them from crossing barriers and increase protein recognition [19], as mentioned previously. Therefore, the use of a PEG coating on sub-10 nm NPs could annihilate their physical advantages. Moreover, PEG's ability to prevent protein adsorption is diminished by its in vivo degradation by reactive oxygen species or transition metals found in most tissues [28]. Lastly, studies carried out on gold nanoparticles [29] indicate that PEG triggers an immune response after a second dose which reduces its circulation time by 20%. Therefore, it is of interest to find an alternative to PEG, in terms of conservation of the NP size and avoidance of toxic hazards. Zwitterions (ZWs) seem to be a good substitute for surface functionalization of nanoparticles for in vivo applications, as reported in previous studies on soft nanoparticles [30,31] and hard nanoparticles such as silica NPs [32–35], gold NPs [29,36,37], or other NPs [38–40]. Indeed, they are globally neutral, highly hydrophilic molecules and should efficiently inhibit protein adsorption. Furthermore, as cell membranes have a zwitterionic nature themselves [41], functionalized nanoparticles should be highly biocompatible and should not trigger any immune response [29]. Moreover, the use of zwitterions paves the way for a huge diversity of molecules, from amino acids to polymers, that can be developed for a specific biological application. To date, there have been no reports of zwitterions being used to functionalize PLNPs or ZGO NPs.

Driven by our interest in developing stealth NPs, a small sulfobetaine-derived zwitterion (MW 330 Da) was synthesized and successfully grafted to the ZGO in a single step and in aqueous conditions. The knowledge our laboratory possesses on those NPs allowed us to use them as a model for NPs, in order to compare them with previously published coatings [18,42]. This new way of ZGO coating using a small and non-toxic zwitterionic molecule [43–48] showed better aqueous stability and protein repulsion, and a similar circulation time in mice to their larger PEGylated counterparts. This is the first time PLNPs and ZGO NPs in particular have been coated with ZW molecules, which opens the path for new coating strategies.

## 2. Materials and Methods

Chemicals were obtained from Sigma-Aldrich (Saint Quentin Fallavier, France), solvents from Fluka, and salts from Alfa-Aesar (Haverhill, MA, USA). Alpha-methoxy-omega-

N-hydroxysuccinimide poly(ethylene glycol) PEG MW 5.000 Dalton was bought from Iris Biotech GmbH (Marktredwitz, Germany). Water refers to Millipore water.

### 2.1. Synthesis of Chromium-Doped Zinc Gallate Nanoparticles

Chromium-doped zinc gallate nanoparticles (ZGOs) were synthesized by hydrothermal method and calcination in air, as previously reported [5]. First, gallium nitrate was formed by reacting 1.67 g of gallium oxide with 10 mL concentrated nitric acid (35 wt%) under hydrothermal conditions at 150 °C overnight. Then, a mixture of 16 mg of chromium nitrate and 2.67 g of zinc nitrate in 10 mL of water was added to the previous solution of gallium nitrate under vigorous stirring. The resulting solution was adjusted to pH 8 with an ammonia solution (30 wt%), stirred for 3 h at room temperature, and transferred into a 40 mL Teflon-lined stainless-steel autoclave for a 24 h heat treatment at 120 °C. The resulting compound was washed with water and twice with ethanol before drying under vacuum at 60 °C for 2 h. The white powder was finally calcinated at 750 °C for 5 h. Hydroxylation was performed by acidic wet grinding of the powder (500 mg) for 15 min with a mortar and pestle in 50 mL of 5 mM HCl solution, and overnight vigorous stirring of the resulting suspension at room temperature. Nanoparticles with a diameter of 90 nm were selected from the whole polydisperse colloidal suspension by centrifugation at 4600 rpm for 12 min. The supernatants were gathered and concentrated to a final 5 mg/mL suspension.

### 2.2. Nanoparticle Characterizations

Five-minute acquisitions of the luminescence of nanoparticles were recorded on an Optima (Biospace, Nesles la vallée, France) camera after a two-minute excitation under UV light (365 nm) or under an LED lamp with a 515 nm high-pass filter. Dynamic Light Scattering (DLS) was used to determine the hydrodynamic diameter (20 µL of washed nanoparticle suspension diluted in 500 µL of aqueous solution of interest) and $\zeta$-potential measurements (20 µL of washed nanoparticle suspension diluted in 800 µL of 20 mM NaCl, 2.5 mS/cm) were obtained with a Zetasizer Nano ZS (Malvern Instruments, Palaiseau, France). ATR FT-IR spectra were acquired on an IRAffinity-1 (Shimadzu, Bd Salvador Allende Noisiel, France). NMR experiments were performed using a Bruker 400 MHz spectrometer in DMSO-d6 (Brüker AVANCE III Spectrophotometer, Palaiseau, France). TGA experiments were conducted on a TGA/DSC 1 (Mettler Toledo S.A., Barcelona, Spain) between 25 and 800 °C (scan rate: 10 K/min) by using 70 µL alumina pans. Each TGA blank curve (empty pan) was subtracted from the TGA sample curve to obtain reliable baselines. The thus-obtained final curve was then normalized to the sample mass.

### 2.3. Synthesis of SBS and Coating of ZGO NPs

The synthesis of 3-{[dimethyl(3-trimethoxysilyl)propyl]ammonio}propane-1-sulfonate (SBS) was adapted from the literature [32]. Briefly, 0.89 g of propane sultone was placed under argon in 7 mL of dry acetone. A total of 1.5 g of (N,N-dimethyl-3-aminopropyl) trimethoxysilane (DAPTMS) was then added and the resulting solution was stirred for 4 h at ambient temperature. The white solid was collected by vacuum filtration, washed twice with acetone by centrifugation (13,400 rpm, 10 min), dried under vacuum at 60 °C overnight, and stored under argon. For the surface functionalization, 3 mg of ZGO was first washed with water by centrifugation (13,400 rpm, 10 min) and suspended in 1.2 mL of 5 mM NaOH and 34.1 mg of the previously synthetized SBS solubilized in 150 µL of 5 mM NaOH were added. After sonication (30 s), the suspension was stirred for 6 h at 80 °C. The resulting ZGO-SBS nanoparticles were washed twice with water by centrifugation (13,400 rpm, 10 min) before being dried for storage before use.

### 2.4. Preparation of PEG-Coated Nanoparticles

The PEGylation of ZGO nanoparticles was performed in organic solvent by a two-step process already described by our group in the literature [5]. Briefly, 5 mg of ZGO was washed with water and twice with DMF by centrifugation (13,400 rpm, 10 min) before being

suspended in 2 mL of DMF. After the addition of 20 µL of (3-aminopropyl)-triethoxysilane (APTES), the suspension was sonicated for 30 s and stirred for 6 h at ambient temperature. The resulting particles were washed twice with DMF by centrifugation (13,400 rpm, 10 min) and resuspended in 2 mL of DMF. Finally, 50 mg of NHS-activated PEG (MeO-PEG$_{5kDa}$-NHS) was added and the suspension was stirred overnight at 90 °C, then washed twice with DMF by centrifugation (13,400 rpm, 10 min) before being dried for storage before use.

### 2.5. Stability of the Nanoparticles in Different Media

A 2 mg/mL suspension of the different NPs in different media was followed by DLS until aggregation. The media used were water, 32.5 mg/mL and 65 mg/mL human serum albumin, and 50% *w/w* mice serum in 5% *w/w* glucose solution.

### 2.6. Quantification of Adsorbed Proteins through Bradford Assay

The commercial Bradford assay solution (Bio-Rad) was diluted 5 times in PBS and filtered using a 0.2 µm Whatman paper filter. ZGO NPs were incubated at a 2 mg/mL concentration in 50% mice serum in 5% glucose at 37 °C for 2 h. Nanoparticles were washed from unbound proteins by several centrifugation steps (13,400 rpm for 15 min at room temperature). The absence of unbound proteins in the last supernatant was verified using the Bradford assay. A total of 10 µL of a 1 mg/mL suspension of washed nanoparticles in 5% glucose was transferred to a 96-well plate (6 wells per sample). Next, 200 µL of Coomassie blue dye reagent (Bio-Rad) was added to each well, and the plate was incubated at 37 °C for 10 min. Absorbance at 595 nm was measured using a plate reader (Tecan Infinite F200Pro, Mannedorf, Switzerland).

### 2.7. In Vivo Biodistribution Studies

Balb/cJRj female mice, age-matched (eight weeks of age, adult) and weight-matched (18–22 g), were purchased from Janvier labs. A total of 5 mg of dried ZGO-SBS or ZGO-PEG nanoparticles were re-dispersed in 800 µL 5% glucose. A total of 200 µL (equivalent to 1.25 mg of nanoparticles) was collected with a syringe and excited for 2 min under a 365 nm UV lamp. Three mice were chemically anesthetized with a mix of rompun and ketamine. The nanoparticles were then injected. During re-excitations and acquisitions, mice were placed on their back and kept asleep with isoflurane. Luminescence was acquired with an Optima camera (Biospace) at different post-injection times for 10 min. For acquisition times longer than 1 h after injection, nanoparticles were re-excited for 2 min under a 515 nm lamp before acquisition.

## 3. Results and Discussion

### 3.1. Preparation of Zwitterionic and PEGylated Zinc Gallate Nanoparticles

The chromium-doped zinc gallate nanoparticles (ZnGa$_{1.995}$Cr$_{0.005}$O$_4$, referred to as ZGOs) were synthesized by a two-step method combining a hydrothermal treatment and a 5 h long calcination at 750 °C. In agreement with previous results [5,18], the obtained nanoparticles emit long-lasting persistent luminescence after excitation in the UV or visible (550 nm) range, as reported in Figure S1. Monodisperse ZGOs of around 90 nm (hydrodynamic diameter) were obtained by acidic hydroxylation of nanoparticles and extraction by selective sedimentation. However, further functionalization is needed since the as-prepared hydroxylated ZGO nanoparticles are known to be rapidly cleared from the blood system in vivo.

PEGylation, the surface functionalization usually performed on nanoparticles and already known to be efficient for in vivo applications [5,18], was used as a benchmark. For this purpose, aminopropyltriethoxysilane (APTES) was first grafted on the ZGO surface during an already optimized 6 h reaction in organic solvent (N,N-dimethylformamide, DMF). Then, the amine-covered nanoparticles were reacted with a solution of N-hydroxysuccinimide activated 5 kDa polyethylene glycol in the same organic solvent (DMF) to give PEGylated ZGO, referred to as ZGO-PEG, as presented in Figure 1.

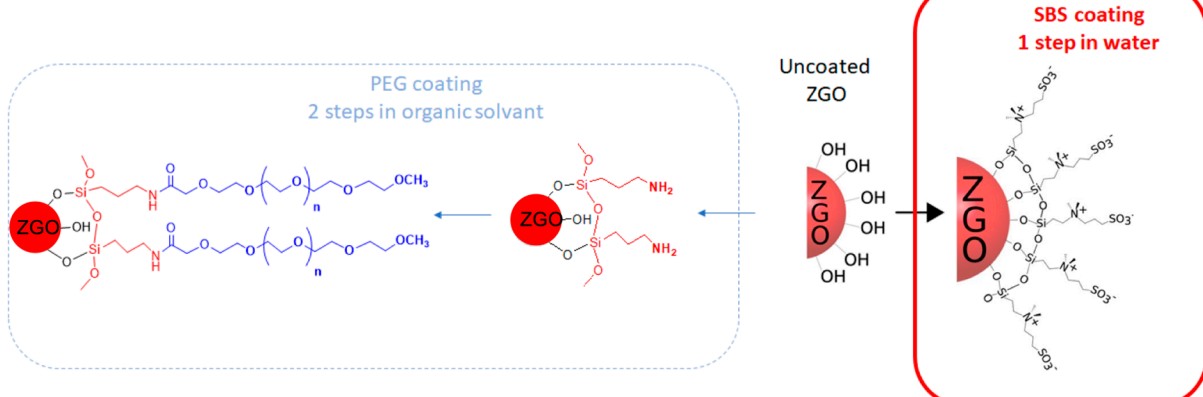

**Figure 1.** Functionalization of ZGO NPs by the SBS zwitterion in one step in water (red, right) and with PEG in two steps in organic solvent (blue, left).

Functionalization by a small zwitterion may help circumvent the issues encountered in vivo with PEGylation. We focused on the functionalization of ZGO nanoparticles by a zwitterion derived from the ring opening of propane sultone on (N,N-dimethyl-3-aminopropyltrimethoxysilane, DAPTMS), a tertiary aminopropyltrialkylsilane close to the already used APTES. The resulting 3-{[dimethyl(3-trimethoxysilyl)propyl]ammonio}propane-1-sulfonate (later named SBS) is highly soluble in water. The sultone derivative was chosen over the lactone one because of the larger pH range in which the zwitterion exists and the better protein repulsion of sulfonates than carboxylates [49]. FT-IR and $^1$H-NMR data of the synthetized ligand can be found in the electronic Supplemental Information.

Functionalization of ZGO by SBS was developed keeping in mind the fixation of APTES on ZGO and the grafting of SBS on silica nanoparticles available in the literature [32].

The influence of several parameters such as the reaction time, the SBS ratio, and the pH were studied (Tables S1 and S2). The best result was obtained when 11 weight equivalents (w.e.) of SBS were added to a 2.2 mg/mL ZGO suspension in 5 mM NaOH and the solution was stirred for 6 h at 80 °C. After washing with water, the obtained nanoparticles show a slight increase of 5 nm in hydrodynamic diameter, which is not surprising considering the small size of the grafted molecule, in opposition to the 95 nm increase observed after PEGylation. Zeta potential measurement is the best indicator of successful functionalization. It dropped from +11 mV for ZGO-OH to −1 mV for ZGO-SBS; hydroxyl groups are therefore correctly neutralized thanks to the zwitterion grafting. These conditions were obtained after optimization of reaction time and SBS amount. Indeed, as seen in Table S1, a shorter reaction time (3 h) does not lead to a neutral surface, meaning that surface zwitteration was not finished. On the other hand, leaving the reaction overnight leads to an increase in hydrodynamic diameter due to the growth of several polymerized SBS layers and colloidal instability. The same happens when putting too much SBS, as reported in Table S2, and 11 weight equivalents of SBS seems to be the best compromise between quantity and stability. Zwitteration leads to neutral nanoparticles half the size of PEGylated ones, as summarized in Table 1. Moreover, these nanoparticles were stable for over a month in water while non-functionalized ZGO and ZGO-PEG lasted only three days and ZGO-PEG four days before aggregation (Figure S2).

**Table 1.** DLS characterizations hydrodynamic diameter (HD) (20 µL of washed nanoparticle suspension diluted in 500 µL water) and zeta potential (ZP) of functionalized ZGO nanoparticles.

| Type | Solvent's Reaction | HD (nm) (PDI) | ZP (mv) |
|---|---|---|---|
| ZGO-OH | Water | 87 ± 2 [0.12] | +11 |
| ZGO-PEG | DMF | 178 ± 4 [0.08] | −4 |
| ZGO-SBS | Water | 92 ± 2 [0.14] | −1 |

IR spectroscopy is the quickest way to check whether a ligand is present or not, its integrity, and how it is linked to the surface. As seen in Figure S3, ZGO-SBS nanoparticles share some characteristic bands with SBS, such as $CH_2$ stretching at 2950 cm$^{-1}$, C-N stretching of the amine at 1205 cm$^{-1}$, and S=O stretching of the sulfonate group at 1179 cm$^{-1}$. However, the characteristic bands of the methoxy-silane groups at 2840 and 1070 cm$^{-1}$ are no longer visible and are replaced by a large Si-O-Si stretching band between 1000 and 1100 cm$^{-1}$, attesting that hydrolysis and reticulation between the silane groups have occurred. This is consistent with the fact that the fixation of alkoxysilanes at high temperatures should lead to reticulated silanes on the surface and enhanced stability [32].

Thermogravimetric analysis was used to quantify the amount of ligand fixed on the surface. First of all, no weight loss is observed for non-functionalized zinc gallate below 800 °C, as seen in Figure 2. For functionalized nanoparticles, above 150 °C heating leads to the decomposition of the organic part of the sample so that the weight loss can directly be linked to the amount of ligand present on the ZGO surface. A total of 21.0 ± 0.5 μg of SBS was fixed per milligram of ZGO, corresponding to 73 ± 2 nmol/mg or 3.9 ± 0.1 SBS/nm$^2$. If we compare it to the density of oxygen atoms at the surface of ZGO (23 atoms/nm$^2$ calculated from the lattice constant) [50] and take into account the silane reticulation, we can say that the surface is densely functionalized. This density is close to what is obtained with the APTES functionalization on ZGO (3 APTES/nm$^2$ for a 3 h reaction) [12]. Moreover, it is higher than what was obtained on silica nanoparticles in the literature (1.0 SBS/nm$^2$) [32] which were already stable in bovine serum. According to Figure 2, its density is higher than what is obtained with PEG on ZGO nanoparticles (112 ± 1 μg of PEG per milligram of ZGO corresponding to 1.19 ± 0.01 PEG/nm$^2$).

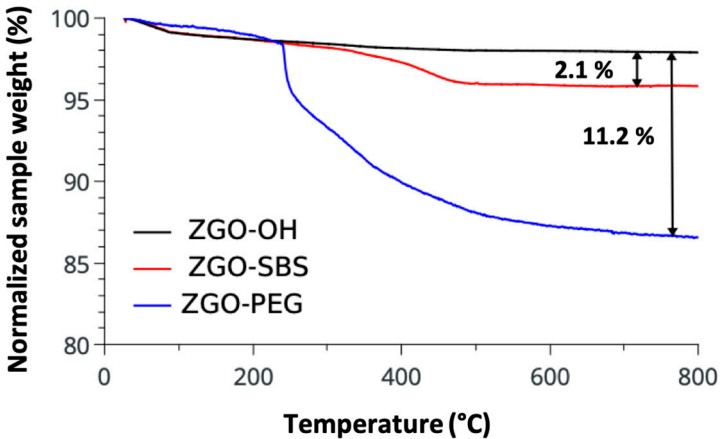

**Figure 2.** Thermogravimetric curves of ZGO NPs before and after functionalization.

We then evaluated the influence of the different coating on the persistent luminescence signal, which mainly originates from defects around the chromium coming from zinc and gallium site inversions and from oxygen vacancies [51]. Persistent luminescence decay after UV excitation is slightly longer after functionalization, with a characteristic time of around 160 s after functionalization instead of 123 s without any coating (Figure S4). The overall intensity also varies: it is multiplied by 1.9 for ZGO-PEG when comparing the same weight of core zinc gallate. This ratio goes up to 2.5 for ZGO-SBS, indicating that reticulation could impose more constraints on the lattice structure. The restoration of traps deactivated during hydroxylation or the creation of new defects on the surface could be the origin of an increased luminescence intensity and time decay after UV excitation. However, the luminescence mechanism and thus the effect of functionalization are different after visible light excitation. According to Figure S4, luminescence decays and intensities are similar before and after functionalization. This suggests that defects created during functionalization might be substitutions in the surface lattice rather than around interstitial chromium. As previously reported [5], functionalization of ZGO nanoparticles does not

prevent UV excitation nor visible re-excitation, making them still applicable for in vivo imaging. Moreover, far from being a disadvantage, SBS is even beneficial for imaging after UV excitation, the persistent luminescence being stronger and lasting longer than for ZGO-OH.

### 3.2. In Vitro Protein Corona Studies and In Vivo Behavior

Before evaluating the in vivo efficacy, we had first to make sure that the ZGO-SBS nanoparticles were colloidally stable. Since recognition by opsonins is responsible for the elimination of ZGO nanoparticles from the blood circulation, it was interesting to estimate the strength of the interaction between ZGO and proteins and analyze the influence of the surface state (OH, PEG, and SBS). If they are found to be of the same order as with PEG, the circulation times in vivo might be similar for the two coatings [41].

Albumin is the most abundant protein in serum (around 40 g/L of albumin for a total concentration of approximately 65 g/L in proteins). It is one of the best protein to study the behavior of ZGO alongside serial proteins. ZGO nanoparticles with different coatings were suspended at 2 mg/mL in 65 and 32.5 mg/mL human serum albumin (HSA) solutions in 5% glucose. These two environments were used to mimic a pure serum environment and a blood-like diluted serum environment, the latter being the one usually studied for complex media stability studies. During incubation, a protein corona forms, and its size can be monitored by DLS size measurements and used to estimate the affinity of proteins for the surface of nanoparticles. Figure 3 shows the size evolution of ZGO in albumin solutions during the first three hours. It is to be noted that a longer incubation (experiment conducted over 24 h) does not lead to a further change in hydrodynamic size. Indeed, regardless of the surface state of ZGO, the final size is obtained after an hour. The rapid interaction between nanoparticles and proteins complies with what has been found in the literature [52].

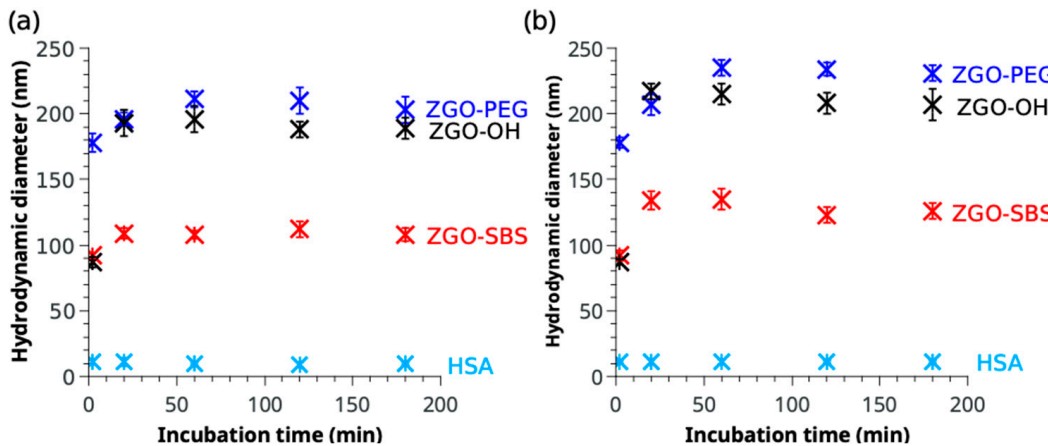

**Figure 3.** Protein corona size studies on different ZGO nanoparticles suspended at 2 mg/mL in human serum albumin (HSA) solutions at (**a**) 32.5 mg/mL and (**b**) 65 mg/mL.

Even though albumin is always introduced in large excess, protein concentration has a strong influence on the final size of the protein coronas. When changing the incubation concentration from 32.5 mg/mL to 65 mg/mL, the final hydrodynamic diameter shows a 20 nm increase corresponding to an additional layer of HSA on the surface (HSA diameter: 10 nm). Thus, a study in 50% diluted serum can help in classifying the affinity of proteins to the different surfaces but not in quantifying the interactions. After incubation with HSA, ZGO-OH led to the thickest protein corona (50 nm thick at 32.5 mg/mL, 60 nm thick at 65 mg/mL), which, if we suppose that both coatings lead to similar colloidal stability, would correspond to around five to six layers of proteins. PEGylation decreases the number of interactions, leading to protein corona thicknesses of 12 and 26 nm (one to three layers of HSA). This can be explained by the hydration of the polymeric chains which render

surface-protein interactions as favorable as water-protein interactions. Further thinning of the corona is achieved by zwitteration, with coronas only 10 and 18.5 nm thick (one to two albumin layers). This result is due to the bigger hydration of the surface due to its zwitterionic nature. ZGO-SBS nanoparticles seem to be the nanoparticles with the least affinity for proteins.

Blood serum is a complex medium that is not only composed of albumin, and incubation in mouse serum will give a better idea of what might happen in vivo. The same monitoring was hence executed in 50% mouse serum diluted in 5% glucose. The results are gathered in Figure 4. First, the three types of nanoparticles can be injected in vivo as none of them aggregate in serum even after a long period of time (even after 24 h, not shown here). However, the interaction between serum and functionalized nanoparticles leads to a corona six to seven times thicker than what was observed with albumin only. Indeed, other proteins of different charges or sizes as well as other molecules can interact more easily with ZGO. Albumin, being charged negatively at physiological pH [52], interacts strongly with the surface of the positively charged ZGO-OH and might be the main component of its corona even in serum conditions, resulting in a similar corona size as in the previous experiment. However, ZGO-SBS nanoparticles remain the smallest nanoparticles and exhibit fewer interactions.

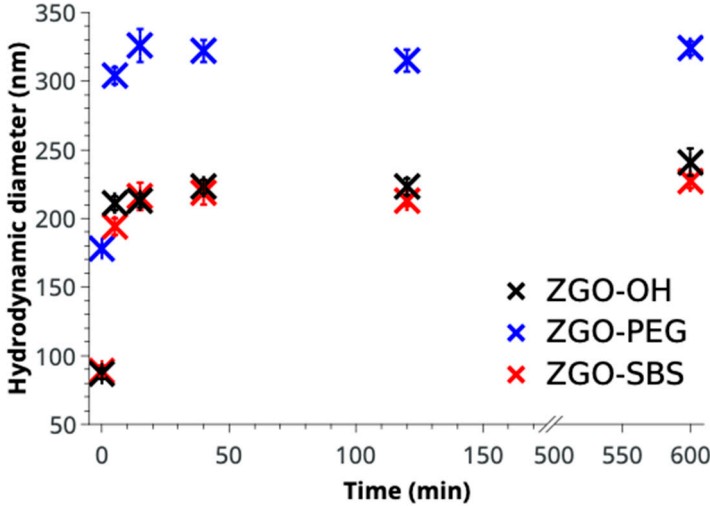

**Figure 4.** Size evolution of the different ZGO nanoparticles at 2 mg/mL in 50% diluted mouse serum in 5% glucose.

The evolution of the hydrodynamic diameter of nanoparticles in serum is the result of several factors other than proteins and is difficult to interpret. The quantification of fixed proteins could be a more adequate way to compare the efficiency of functionalization. To this extent, a Bradford assay was used, coupled with a calibration curve made from a solution of known concentration of albumin (Figure S5). After two hours of incubation of ZGO in 50% diluted mouse serum (more than the time needed for the protein interaction to occur, as seen before), the nanoparticles are centrifuged and washed several times with 5% glucose to remove unbound or loosely bound proteins. The remaining proteins are called "hard corona" and are responsible for the interactions with the environment (the others being loosely bound can be easily exchanged) [52]. Non-incubated ZGO nanoparticles were used as the reference for an absence of bound protein. The results are summarized in Figure 5. The complete removal of unbound proteins is verified by the absence of proteins recovered in the last washing. Unsurprisingly, the quantity of proteins bound on non-functionalized ZGO is bigger ($100 \pm 5$ µg per milligram of ZGO) than on functionalized ones. Zwitteration seems to be slightly more efficient than PEGylation ($68 \pm 7$ µg of protein bound compared to $75 \pm 6$ µg per nanoparticle of core ZGO). Therefore, functionalization with molecules similar to cell membranes, even with a small molecular weight (330 Da),

seems to be a good alternative to the usual PEG (5000 Da) coating to reduce protein adsorption. ZGO-SBS nanoparticles, being resistant to opsonization and smaller than their PEG-coated counterparts, were evaluated for their ability to freely circulate in the blood of mice.

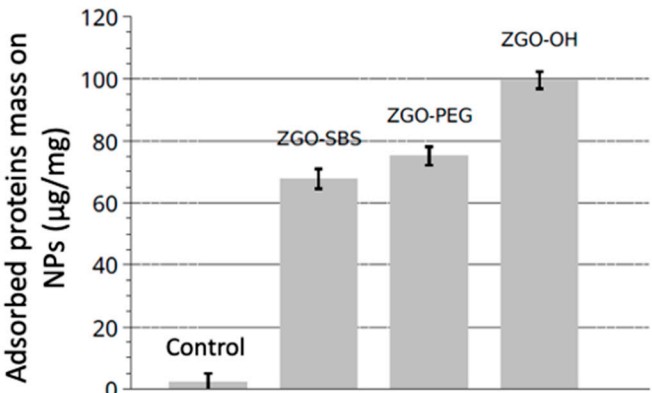

**Figure 5.** Quantification of mouse serum proteins bound on the different ZGO following Bradford assay. The control used is ZGO nanoparticles not incubated with serum. The supernatant from the last washing was also tested to ensure complete removal of unbound proteins had occurred.

Finally, in vivo experiments were performed on healthy mice. ZGO-SBS nanoparticles dispersed into 5% glucose were excited with a UV lamp and then injected into the tail vein and imaged. During the first 60 min, mice were imaged without any further excitation. As seen in Figure 6, contrary to ZGO-OH nanoparticles which are captured in the liver in less than 10 min [18], the signal emitted by the ZGO-SBS nanoparticles is well distributed in all the mice body. Three hours after the injection, mice were re-excited with LED. Again, and contrary to what was previously observed with small magnetic nanoparticles [53], we observe that many NPs continue to freely circulate into the blood vasculature for up to 6 h, as observed with the PEGylated ZGO presented in Figure S6 using higher molecular weight grafted molecules [18,42].

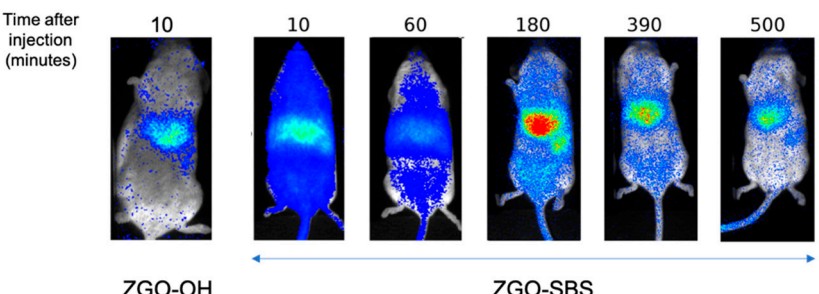

**Figure 6.** In vivo biodistribution in mice of ZGO-SBS nanoparticles (*n* = 3).

## 4. Conclusions

We have developed a new method to functionalize the surface of persistent luminescent nanoparticles in order to obtain stealth NPs. For this purpose, we have replaced the usual PEG (MW = 5000 g/mol) coating realized in two steps in organic solvent with a much smaller (MW = 300 g/mol) zwitterionic molecule in one step in water. Despite a more exposed core, the zwitterionic surface leads to smaller nanoparticles, better luminescence properties, and a more efficient protein-repelling character. These properties allow the zwitterionic nanoparticles to have a similar biodistribution to PEGylated ZGO. The successful development of zwitterionic coating on ZGO model nanoparticles could pave the way for the use of sub-10 nm nanoparticles.

**Supplementary Materials:** The supporting information can be downloaded at: https://www.mdpi.com/article/10.3390/coatings13111913/s1.

**Author Contributions:** Conceptualization, C.R.; methodology, Y.C., J.M. and J.S.; validation, J.L. and G.C.; investigation, D.D. and T.L.; writing-review and editing, T.L., C.R., N.M. and D.S., funding acquisition, C.R., N.M. and D.S. All authors have read and agreed to the published version of the manuscript.

**Funding:** This work was supported by a grant from the French Research Agency (ANR-14-CE08-0016-01).

**Institutional Review Board Statement:** All experiments involving mice were approved by the French *Comité d'éthique en expérimentation animale* N°034 and by the French Ministry of Research APAFIS#8519-20 16090514387844.

**Informed Consent Statement:** Not applicable.

**Data Availability Statement:** The data presented in this study are available on request from the corresponding author.

**Conflicts of Interest:** The authors declare no conflict of interest.

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
