# Peer review of "Zwitterionic Functionalization of Persistent Luminescence Nanoparticles: Physicochemical Characterizations and In Vivo Biodistribution in Mice"

_coatings, doi:10.3390/coatings13111913_

Round 1

Reviewer 1 Report

Comments and Suggestions for Authors

Dr. Richard and co-authors reported the zwitterionic functionalization of persistent luminescence nanoparticles and their biological imaging applications. Based on the experiments and discussion, it can be accepted after addressing the following issues.

1)     For better illustration of the surface functionalization of the nanoparticles, please update the figure 1 to show the different steps of ZGO-OH, ZGO-SBS and ZGO-PEG.

2)     In lines 328-337, the in vivo experiments were performed on healthy mice using ZGO-OH and ZGO-SBS, what are the results for the ZGO-PEG nanoparticles?

3)     The authors focused on the persistent luminescence of the nanoparticles, what are their basic luminescent properties like absorption and luminescence spectra?

Author Response

1) For better illustration of the surface functionalization of the nanoparticles, please update the figure 1 to show the different steps of ZGO-OH, ZGO-SBS and ZGO-PEG.

Following this remark, Figure 1 has been updated to show both surface functionalization.

2) In lines 328-337, the in vivo experiments were performed on healthy mice using ZGO-OH and ZGO-SBS, what are the results for the ZGO-PEG nanoparticles?

Results of the biodistribution of ZGO-PEG NPs are shown in Figure S6. Qualitatively, ZGO-SBS and ZGO-PEG have showed a similar circulation time into the blood stream.

3) The authors focused on the persistent luminescence of the nanoparticles, what are their basic luminescent properties like absorption and luminescence spectra?

The authors would like to thank the referee to have highlighted this lack of information. Both excitation and emission spectra have been added in the ESI (Figure S1).

Reviewer 2 Report

Comments and Suggestions for Authors

The paper deals with an interesting study of zwitterionic luminescence Na-nanoparticles and its characterization and biodistribution. The study is reasonable but full of mistakes and errors, I strongly suggest the authors read the text critically to eliminate numerous mistakes. Some of my specific comments are given hereunder. 

1. The abstract is relatively long and not really focused.

2. Use units throughout the text in a uniform way and according to IUPAC standards, like minutes and min, hours and h etc.

3. Space between units and their values are very strange at several places.

4. Line 105, change the sentence, experiments can not be carried out on NMR machine, rather the machine is used for characterization.

5. Insert “degree” properly, when representing temperature.

6. Line 111, Check the value 1,5 g, are 1 and 5 separate numbers or it is 1.5 g? same is in line 116, and in line 121 (literature5.5 mg). 

7. Line 120, insert hyphen between two and step (twosteps be replaced by two-steps).

8. Several sentences have been started with either a number or citation of Figure and Table. Revise them.

9. Line 156, Check chemical formula of ZGO, for subscript of molar values in the material.

10. Line 175, IUPAC name of the compound is wrong, with several parenthesis.

11. Revise IR to FT-IR and mind the position of “1” in H-NMR.

12. Format the font within the figure and bring it to comparable size with the text.

13.   Check sentence in line 190, starting with symbol followed by first small letter.

14. Line 206-213, the FT-IR data in the text particularly the cm-1, and fragments/groups with respect to subscripts of numbers is not proper (CH2). 

15. Line 220, what is SBS/nm2? Atoms/nm2, APTES/nm2 and so on. Isn't it nm2?

16. Several sentences are not written properly please change them.

17. In conclusion, References are unusual, it must be based on the work presented in the instant paper. Polish this section and make it more concise.

18.   Section Supplementary material, is incomplete.

19. Figure 6 and S5 have same caption, but they are different, for the study of ZGO-SBS, why?  

Based on the suggestions given above the MS needs major revision. 

Comments on the Quality of English Language

Extensive Editing is needed. 

Author Response

1) The abstract is relatively long and not really focused.

The abstract has been modified following this remark.

2-14)

The authors would like to thank the referee to have noticed these mistakes. The manuscript has then been revisited, some of the modifications have been highlighted in yellow.

15) Line 220, what is SBS/nm2? Atoms/nm2, APTES/nm2 and so on. Isn't it nm2?

The authors would like to thank the referee to have noticed these mistakes. It was indeed “nm2”. The manuscript has then been revisited, some of the modifications have been highlighted in yellow.

16) Several sentences are not written properly please change them.

The authors would like to thank the referee to have noticed these mistakes. The manuscript has then been revisited, some of the modifications have been highlighted in yellow.

17) In conclusion, References are unusual, it must be based on the work presented in the instant paper. Polish this section and make it more concise.

It is true that references in the conclusion are unusual but the authors thought that they will bring some perspectives to it. The authors will defer to the publisher’s decision.

18) Section Supplementary material, is incomplete.

A new figure (Figure S1) containing the absorption and emission spectra has been added to the ESI after a remark from Referee 1.

19) Figure 6 and S5 have same caption, but they are different, for the study of ZGO-SBS, why?  

The authors would like to thank the referee to have noticed this mistake. The manuscript has then been revisited, some of the modifications have been highlighted in yellow.

Reviewer 3 Report

Comments and Suggestions for Authors

I read this article carefully. I think this work is interesting and conforms to the scope of the journal. This article can be accepted after major revision. Some comments are as follows:

1. Please mention Figure 1 in the main text.

2. Please check the upper and lower corner markers.

3. The authors had better cite some novel references in this article.

4. How about the cytotoxicity?

5. Please check data carefully to avoid errors.

Author Response

1) Please mention Figure 1 in the main text.

The authors would like to thank the referee to have noticed this mistake. The manuscript has then been revisited, some of the modifications have been highlighted in yellow.

2) Please check the upper and lower corner markers.

The authors would like to thank the referee to have noticed these mistakes. The manuscript has then been revisited, some of the modifications have been highlighted in yellow.

3) The authors had better cite some novel references in this article.

26 new references have been added in the revised version

4) How about the cytotoxicity?

The in vitro and in vivo toxicity of the ZGO NPs has already been evoked in this article (ref 14-18). Some new references have been added concerning the toxicity of SBS (ref 44-49). The authors themselves did not perform any cytotoxicity assessment. The goal of this study was to demonstrate the possibility and the advantages to replace a PEG coating with a zwitterionic one. Considering the existing literature, the authors decided to not add this kind of characterization.

5) Please check data carefully to avoid errors.

The authors would like to thank the referee to have noticed these mistakes. The manuscript has then been revisited, some of the modifications have been highlighted in yellow.

Reviewer 4 Report

Comments and Suggestions for Authors

The results presented in the manuscript of D. Dassonville et al. (Art. No. coatings-2653434) are new and could be interesting for readers. Some negligible improvement could be done.

  1. Figure 1. In my opinion, the functionalization scheme should be more detailed.
  2. Figure 2. Figure caption should be changed to “TG curves of ZGO nanoparticles before and after functionalization”. Moreover, the weight loss marked in Figure should be expressed in the same way. For example, 2.1 % and 11.2 %.
  3. Since authors are talking about “persistent luminescence nanoparticles”, the excitation and emission spectra should be added to the text.
  4. The English should be re-checked.

The manuscript could be suitable for publication in this journal after major revision.

Comments on the Quality of English Language
  1. The English should be re-checked.

Author Response

1) Figure 1. In my opinion, the functionalization scheme should be more detailed.

Following this remark, Figure 1 has been updated to show both surface functionalization.

2) Figure 2. Figure caption should be changed to “TG curves of ZGO nanoparticles before and after functionalization”. Moreover, the weight loss marked in Figure should be expressed in the same way. For example, 2.1 % and 11.2 %.

The authors would like to thank the referee to have noticed these mistakes. Figure 2 has then been updated as demanded.

3) Since authors are talking about “persistent luminescence nanoparticles”, the excitation and emission spectra should be added to the text.

The authors would like to thank the referee to have highlighted this lack of information. Both excitation and emission spectra have been added in the ESI (Figure S1).

4) The English should be re-checked.

The authors would like to thank the referee to have noticed these mistakes. The manuscript has then been revisited, some of the modifications have been highlighted in yellow.

Reviewer 5 Report

Comments and Suggestions for Authors

Dear Authors,

The production of persistent luminescence NPs, and in particular the surface modification of such NPs, is of major importance for the development of novel tools for bioimaging. Therefore, this topic can be of interest for many researchers.

The authors presented a new way for surface modification of ZGO NPs in order to get similar results as, the commonly used, PEGylated NPs, but achieving a smaller size of the shell, which can be important in some cases related with the biodistribution of NPs. The overall work is interesting but lacks on a carefully exposition of the Methodology and discussion of some results. 

More details suggestions and comments are as follow:

Abstract section:

First line of the abstract should be corrected “After excitation in the biological transparency window,” in this work the authors did not excite the NPs in the biological transparency window, they used UV and a visible LED (515 nm), then the NPs emit in the biological transparency window.

Introduction section:

The introduction gives a good background to understand the problematic and the innovation of the presented work.

Line 87: Please defined PLNPs the first time that it is write.

Material and Methods section:

The authors should discriminate which chemical was bought obtained from Sigma-Aldrich, Fluka or Alfa-Aesar.

The synthesis of the NPs is explained in the supplementary information. That is ok as the NPs synthesis was already published elsewhere and is not the focus of this manuscript. Still, the method to produce the NPs and the references should be briefly expose in the Methods section, saying that details of the protocol is on the Supplementary Information.

Line 103: DLS should be defined the first time that it is write.

Line 118: How the ZGO-SBS NPs were washed? Do the authors used centrifugation, for example? If yes, the centrifugation conditions should be mentioned.

Line 121: “literature5”, I believe that this is ref [5], please correct. How are the washing step of NPs performed?

Line 128: “Stability of nanoparticles in different media”. Please rewrite the paragraph for a better understanding.

Line 134: it should be mentioned were the “commercial solution” was bought.

When the authors mention 5% glucose, they should indicate 5% of v/v? w/W?, w/V?

Line 127: the PEG-ZGO NPs were dry after being washed. Was the SBS-ZGO NPs also dry (line 118)?

Please correct in some places the symbol of “degrees” in the temperature.

Results and Discussion section:

The first two paragraphs of the “Results” section are protocol description, it could be summarized in this section to introduce the results and the idea of the work. Moreover, Figure 1 is not mentioned in the text, and it should be.

Did the calcination step of “5-hour-long calcination at 750 °C” provoked NPs aggregation? A SEM or TEM image should be provided to access the size and distribution of NPs.

The authors should add references to support the data assignments of IR and NMR measurements.

In the Methods section it is mentioned that the zeta and DLS measurements were perormed in a solution of 20 mM NaCl. Is this information correct?

Was the functionalized NPs caracterized under different ionic strengths, or just the 20 mM NaCl? As the physiological saline solution (typically 9% NaCl) corresponds to 154 mM. So, why do the authors choose 20 mM of NaCl instead of 154 mM? Moreover, this solution was used just for the DLS and zeta measurements (as indicated in the methods section)? Why DLS and zeta measurements were not performed with the NPs dispersed in water, for instance? (please confirm as in the caption of Table 1 and figure S1 it says that the NPs are dispersed in water!) I believe that there is some confusion in this information. Please comment about this and improve the text in the main manuscript.

It is possible to show me the DLS graph of the ZGO, ZGO-PEG and ZGO-SBS? As suggestion, this information could also be incorporated into the SI, together with Figure S1.

Can you provide an explanation why the ZGO-OH NPs have a zeta potential of + 11mV? What provides the slightly positive charge to the NPs? Then, in line 191, the authors say that the OH- are neutralized…this does not seem correctly explain.

Table 1: in the caption says that the NPs are diluted in water, but in the table there is a column refereeing a solvent, namely for the ZGO-PEG. Was the DLS and zeta potential measure in water of DMF? Please clarify.

Paragraph about FTIR measurements: please write “cm-1” with the “-1” superscript.

Also, regarding FTIR measurements, I was expected to observe a broad band around 3400 cm-1 assigned to OH in the ZGO-OH NPs. This should be then decrease after the functionalization. Have the authors any evidence that the NPs have hydroxyl groups at the surface?

Line 220: please correct and/or clarify, when the authors say 3.9 SBS/nm2, this is 3.9 molecules of SBS per area of the NPs (nm2)? So, how many molecules per NP? How the authors estimated the number of NPs per mg of ZGO powder?

Photoluminescence (PL) spectra of the dry powder of NPs and the corresponding NPs dispersed in solution should be provided, at least at supplementary information as the NPs are already well characterized elsewhere.

PL spectra – Do the surface functionalization has any effect on the PL outcome? For example, PL intensity and/or any shift of wavelength? This is not clear in the main manuscript, and I believe that will help if the authors display the PL spectra.

I suggest that the persistent luminescence spectra, should be in the main manuscript. The persistent luminescence is monitored at what wavelength? This should be mentioned.

Line 245: please correct “reported5”

While presenting and discussion Figure 6, Figure S5 should be referred.

Comments on the Quality of English Language

The English to understand the work is good and just a few minor errors where found.

Author Response

The authors would like to thank the reviewer for his/her time and inputs to this article. The answers to your questions have been written in the attached word file. Modifications have been brought to the manuscript and are highlighted in yellow.

Round 2

Reviewer 1 Report

Comments and Suggestions for Authors

It can be accepted.

Author Response

The authors would like to thank the reviewer for his/her time and inputs to this article.

Reviewer 2 Report

Comments and Suggestions for Authors

Authors have made all changed appropriately, however I do not agree with the conclusion. I recommend authors to delete references from this section and further improve it.  

Comments on the Quality of English Language

Minor editing in English language is required. 

Author Response

The authors would like to thank the reviewer for his/her time and inputs to this article. 

As said previously, references in the conclusion are unusual but not forbidden. Since none of the 4 other reviewers have declared any issue with the conclusion, we decided to stay on our ground and let the editors decide what to do.

Reviewer 3 Report

Comments and Suggestions for Authors

I read this revised manuscriptcarefully. I think the authors response most comments, so this version can be accepted.

Author Response

(The authors gave the same response as above.)

Reviewer 4 Report

Comments and Suggestions for Authors

Since authors addressed carefully all my comments, the manuscript in my opinion now is suitable for publication.

Author Response

(The authors gave the same response as above.)
